# Transcriptional and Epigenomic Markers of the Arterial-Venous and Micro/Macro-Vascular Endothelial Heterogeneity within the Umbilical-Placental Bed

**DOI:** 10.3390/ijms231911873

**Published:** 2022-10-06

**Authors:** German A. Arenas, Nicolas Santander, Bernardo J. Krause

**Affiliations:** 1Laboratorio de Neurobiología, Facultad de Ciencias Biológicas, Pontificia Universidad Católica de Chile, Santiago 8331150, Chile; 2Instituto de Ciencias de la Salud, Universidad de O’Higgins, Rancagua 8370993, Chile

**Keywords:** endothelial, placenta, umbilical, epigenetics, phenotypic heterogeneity

## Abstract

Umbilical and placental vessels and endothelial cells (EC) are common models to study placental function and vascular programming. Arterio-venous differences are present in the umbilical endothelium; however, the heterogeneity of small placental vessels and the expression of potential micro- vs. macro-vascular (MMV) markers are poorly described. Here, we performed a meta-analysis of transcriptomic and DNA methylation data from placental and umbilical EC. Expression and methylation profiles were compared using hierarchical clustering, dimensionality reduction (i.e., tSNE, MDS, and PHATE), and enrichment analysis to determine the occurrence of arterio-venous (AVH) and micro-macro heterogeneity (MMH). CpG sites correlated with gene expression of transcriptional markers of MMH and AVH were selected by Lasso regression and used for EC discrimination. General transcriptional profile resulted in clear segregation of EC by their specific origin. MM and AVH grouping were also observed when microvascular markers were applied. Altogether, this meta-analysis provides cogent evidence regarding the transcriptional and epigenomic profiles that differentiate among EC, proposing novel markers to define phenotypes based on MM levels.

## 1. Introduction

The placental endothelium is multifunctional and heterogeneous, playing a key role in fetal physiology [1]. To fulfill this role, endothelial cells (EC) in the placenta show marked functional diversity, which occurs early in vascular development and allows it to cope with specific requirements in a time- and site-specific manner [2]. However, the extent to which these differences are defined by the arterio-venous nature or by their corresponding vascular level (micro- vs. macrovascular) remains elusive.

The arterio-venous nature of EC is defined early during development, an effect mainly related to the exposure of the primordial endothelium to gradients of shear stress and vascular growth factors [3]. These effects result in a differential expression of genes between the endothelium of arteries and veins, which would be consistent across the vasculature and can be found even at the level of capillaries [4,5]. These phenotypes are also controlled by the epigenetic landscape, affecting promoters and enhancers, which finely tunes the expression of arterio-venous genes [6,7]. By leveraging multiple transcriptomic datasets, we have recently shown that arterio-venous heterogeneity is preserved in umbilical endothelium in vitro [8]. This transcriptional heterogeneity involves several transcripts and non-coding RNA, affecting key vascular functions. Similarly, the endothelium of the microvasculature—which includes all “small” vessels: arterioles, capillaries, and venules—shows remarkable molecular heterogeneity among cells of different types and in different organs [9,10,11]; however, in contrast to arterial-venous phenotypes, there is no consensus for microvasculature markers [12].

The phenotypical endothelial variability in the placenta has been partially explored. For instance, studies concerning this organ use either EC from the umbilical cord or whole cotyledons [13,14,15]. The latter preparation is expected to contain mainly microvessels, but it cannot distinguish the heterogeneity of EC within the villi, which is expected to have functional correlates [16]. Furthermore, if microvascular differences are based on the arterio-venous identity has not been completely addressed (reviewed in [1,2]). In this study, we compared transcriptomic and epigenomic profiles from umbilical and villous arteries and veins to identify molecular markers of micro- and macro-vascular heterogeneity and to uncover potential functional differences due to this heterogeneity. The results derived from this analysis may help in future research with single-cell technologies to guide the identification of endothelial populations occurring within placental tissue.

## 2. Results

### 2.1. Transcriptional Profiling of Umbilical and Placental Endothelial Cells

Analysis of transcriptomic data from umbilical-placental EC showed that 17,280 transcripts were commonly expressed in PLAEC, HUAEC, PLVEC, and HUVEC. Hierarchical clustering of all samples resulted in two main clusters differentiating between umbilical (HUAEC and HUVEC) and placental (PLAEC and PLVEC) EC, followed by a clear clustering of arterial and venous samples, more consistent in microvascular samples than in umbilical ones (Figure 1A). To visualize variability in endothelial molecular signatures, three different dimensionality reduction analyses were performed. MDS (Figure 1B) and t-SNE (Figure 1C) showed that EC grouped according to their vascular bed of origin (i.e., HUAEC, PLAEC, PLVEC, HUVEC), with high proximity between umbilical and placental EC. To determine relative distances among samples, PHATE analysis was applied, resulting in compact grouping among samples from a defined vascular bed, with high proximity between HUAEC and HUVEC compared to other pairs (Figure 1D).

### 2.2. Discovering Microvascular-Specific Transcripts

Considering that umbilical vs. chorionic origin was the main trait explaining transcriptomic differences, we aimed to identify putative general micro vs. macrovascular endothelium markers. EC from the human aorta (HAoEC), saphenous vein (HSVEC), brain microvasculature (HMEC-brain), lung microvasculature (HMEC-lung), HUVEC, PLAEC, and PLVEC were initially compared (discovery datasets). These comparisons showed that the organ of origin was the main factor of variation (76%), followed by vascular level (24%) (Figure 2A,B). Micro vs. macrovascular comparison resulted in 4142 differentially expressed genes (DEGs) (Figure 2C, Appendix A). Using these DEGs to cluster all samples led to a partial clustering according to the level of origin (Figure 2D). Conversely, umbilical vs. placental EC comparison (validation dataset) resulted in 12,842 DEGs (Appendix A), a profile with a limited concordance with DEGs found in the discovery dataset when up- and downregulated genes were compared (Figure 2D). Based on the top 50 DEG found in both discovery and validation datasets, clustering and PHATE analysis were performed for umbilical and placental EC, improving the clustering and grouping observed in Figure 1, but with maintained differentiation between arterial and venous cells (Figure 2E,F). Furthermore, Lasso-selected transcripts from umbilical-placental comparison resulted in 14 genes that showed high predictive value for micro/microvascular discrimination for fetal and adult EC (Appendix A). Gene Ontology, KEGG, and Reactome pathway analysis revealed that genes differentially regulated between umbilical and placental EC were enriched in terms and pathways related to development, metabolic processes, and signaling pathways (e.g., cAMP, NF-κB). “Regulation of nitrogen compound metabolic process” term showed the highest number of matching genes while “Anatomical development process” showed the lowest adjusted *p*-value (Figure 2F).

### 2.3. Transcriptional Markers of Arterial-Venous Identity for Umbilical and Placental Vessels

To address the preserved segregation between arterial and venous EC, we aimed to determine if validated arterio-venous markers can be differentially expressed among umbilical and placental EC. Artery–vein comparison resulted in 1274 DEG (Appendix A), a profile that modified the relative position of HUAEC and PLAEC in PHATE analysis (Figure 3A). Gene Ontology and pathways (KEGG, Reactome) analysis showed that arterio-venous genes differentially regulated were enriched in terms and pathways related to the circulatory system and cellular responses (i.e., to oxygen, EC migration). “Cellular response to oxygen-containing compound” showed the highest number of matching genes, while “Regulation of actin cytoskeleton organization” showed the lowest adjusted *p*-value (Figure 3B).

Analysis of cross-platform validated arterial-venous 44 DEGs [8] resulted in a clear clustering of HUAEC and PLAEC vs. HUVEC and PLVEC (Figure 3C). To obtain a reduced number of the most influential arterio-venous transcriptional markers, Lasso analysis was applied, retrieving 9 DEGs highly correlated with arterio-venous dichotomy (Figure 3D). These were further validated by logistic regression and showed excellent discriminating ability (Figure 3D,E).

### 2.4. DNA Methylation Profiling of Fetal and Placental EC

We analyzed genome-wide DNA methylation profiles to determine if phenotypic heterogeneity in the umbilical-placental vascular beds was associated with different epigenomic patterns. No epigenomic data from HUAEC was found, therefore, we used epigenomic profiles from pulmonary artery EC to maintain the representation of large arteries. Principal component analysis showed that micro-macrovascular nature was the main source of variability in DNA methylation in these datasets, followed by arterio-venous nature (Figure 4A). Similarly, differentially methylated probes (DMPs) and regions (DMRs) clustered samples according to those variables (Figure 4B,C). PHATE dimensionality reduction showed that a closer relationship occurred between macro and microvascular EC pairs than between arterial or venous pairs (Figure 4D).

Micro vs. macrovascular comparison resulted in 40,110 DMP (FDR <0.05, absolute delta beta >0.2), of which 24,882 occurred in gene contexts (i.e., non-intergenic regions) involving 9660 different genes (Appendix A). Conversely, 2128 DMR were detected based on no beta differences (FDR < 0.05), which were reduced to 590 when beta differences <5% were excluded (Appendix A). Epigenomic-specific functional enrichment analysis showed differential methylation among genes involved in immunoglobulins production, Rho GTPases, and adherens junctions (Figure 4E). Strikingly, we observed differential methylation in sequences associated with the binding of transcription factors canonically involved in vascular development, such as SMADs, GATAs, and KLFs (Figure 4F,G).

Arterial vs. venous EC comparison resulted 16,552 DMP (FDR < 0.05, absolute delta beta > 0.2), of which 11,329 were in gene contexts involving 5694 genes (Appendix A) (Figure 5A). Conversely, 1839 DMRs were detected based on no beta differences (FDR < 0.05), which were reduced to 493 when beta differences < 5% were excluded (Appendix A) (Figure 5B). Epigenomic-specific functional enrichment analysis showed differential methylation among genes involved in multi-organism localization, response to mechanical stimulus, vasculogenesis, hemostasis, and other biological processes (Figure 5C), as well as several regions related to transcriptional factors such as GATA, ESR, NFE2L2, KLF, SP1, and YAP/TAZ (Figure 5D,E).

### 2.5. Validation of Epigenetic Markers for Fetal, Umbilical, and Placental EC Heterogeneity

Next, we sought to uncover epigenetic markers associated with distinct vascular natures. To address this aim, the correlation between gene expression and methylation status in the micro- vs. macro-vasculature, to predict putative epigenetic regulation of transcriptional activity was explored using transcriptional and epigenomic data coming from the same set of samples. A total of 729 probes lay within genes coding for the proposed microvascular markers; most of these were not correlated with changes in gene expression nor were differentially methylated. Only 11 probes showed both differential methylation and correlation with gene expression (Figure 6, Table 1). These were tested as potential markers by Lasso analysis, resulting in 6 probes with a high predictive potential for microvascular EC (Figure 6B) that was further confirmed by logistic regression (Figure 6C).

Conversely, a total of 1186 probes lay within genes coding for arterio-venous markers, and nearly 10% of them showed a correlation with gene expression (Figure 6D, Table 2). A total of 54 DMP that correlated with gene expression were tested as markers by Lasso analysis, resulting in 9 probes with high predictive power for differentiating the arterio-venous nature (Figure 6E), confirmed further by logistic regression (Figure 6F).

## 3. Discussion

This analysis aimed to identify epigenomic markers related to EC heterogeneity and correlate them with transcriptional markers of arterial-venous and micro-macrovascular phenotypes. General transcriptional profile resulted in clear segregation of EC by their specific origin (micro- and macro-vasculature and arterio-venous nature), traits that were maintained even after filtering by microvascular or arterial-venous markers. Similar behavior was observed when the epigenomic profile of PLAEC, PLVEC, and HUVEC was compared. The selection of epigenomic markers with transcriptional correlation resulted in six microvascular and eight arterial DNA methylation sites with high predictive value for those traits, and applicable also for non-umbilical and placental vessels. Altogether, these data provide new insights on epigenomic and transcriptional markers of micro-macrovascular level and arterio-venous nature as new tools to identify and classify different types of EC within the umbilical and placental bed.

Placental vascular development shares its origins and processes with systemic circulation, which results from the migration and differentiation of endothelial progenitors coming from the yolk sac [17]. Early vascular development has long-lasting effects on arterial-venous fate, leading to the expression of ephrin-B2 in arteries, and Eph-B4 in veins, both in fetal [4,18] and adult [4,19] blood vessels. These markers are differentially regulated by Notch (i.e., ephrin-B2) [20] and Nr2f2/COUP-TFII (i.e., Eph-B4) [21,22]. Previous studies have found these specific markers in umbilical and placental endothelium in situ and in vitro [13,23,24], supporting the presence of arterial-venous phenotypes in this vascular bed. However, studies with a reduced representation of endothelial diversity, particularly HUAEC [1,25], and the assumption of micro vs. macro-vascular dichotomy as an exclusive factor [25,26] have limited the characterization of endothelial heterogeneity within the umbilical-placental circuit. To tackle this gap, this study characterized the transcriptional profile of HUAEC, PLAEC, PLVEC, and HUVEC by gathering 84 independent samples, to deliver a comprehensive analysis of the transcriptional phenotypes of these EC by complementing with epigenomic data. Dimensionality reduction and clustering analysis resulted in four well-defined phenotypes, which included the arterial-venous dichotomy as a factor. Furthermore, the application of previously validated markers confirmed a differential expression of arterial-venous transcripts in both umbilical and placental EC. Previous studies on PLAEC, PLVEC, and HUVEC suggest that placental microvascular EC have substantial transcriptional and epigenomic differences [15]; however, those data do not address the contribution of the arterial-venous phenotypes to those differences, as has been found in HUAEC and HUVEC. Altogether, these results strongly support that EC within the umbilical-placental circuit express arterial-venous markers, independently of their location within large or small vessels, but in concordance with their connection with the fetal systemic circulation.

In addition to exploring arterial-venous markers, we studied differences between micro- and micro-vasculature, an area of growing interest [12]. These two levels are commonly described in anatomical terms, specifically size. However, vessel diameter is highly heterogeneous along the placental vasculature, and this is especially true in the villous blood vessels [16]. Here, capillaries, arterioles, and venules can be confounded anatomically and might be part of a continuum along the blood vessels at the villi. In addition, extensive molecular heterogeneity has been recognized in multiple vascular beds throughout the body in mice [9,10,11]. However, existing single-cell transcriptomic datasets lack enough true EC to capture the true extent of endothelial differences in the placenta [27,28]. Conversely, articles providing data about human placenta microvascular EC (HPMEC) are based on protocols that collect simultaneously both arterial and vein microvascular cells (minced villi enzymatic digestion), with no defined markers allowing to identity them as microvascular cells, and controls showing the expression of arterial-venous markers [26,29]. Moreover, reports from the same research group that has applied both complete cotyledon perfusion (starting from the chorionic artery) [23] and separated chorionic artery (PLAEC) and vein (PLVEC) perfusion [14] show comparable phenotypes between the proposed placental microvascular endothelial cells and PLVEC. In this regard, our results showed striking differences between PLAEC and PLVEC, as has been previously described [14,15], calling for attention when villi endothelium is studied as a single phenotype and asking for further studies clearly addressing the vascular level at which endothelial cells are isolated. The identified markers of the placental vasculature in this study may help to solve this issue, with potential applications also within fetal microvascular, or small vessels, phenotypes as suggested by Lasso and logistic regression analysis. The markers identified here will be useful for upcoming algorithms aimed to recognize micro- and macro-vascular EC populations in single cells analysis. In addition, probes selected by Lasso fall within loci of genes involved in several processes important for endothelial cell function, including directed cell migration and solute transport. This suggests that these genes may represent relevant functional differences between endothelial cells of distinct origin.

Compelling data show HUAEC and HUVEC phenotypes are maintained in vitro by epigenetic and post-translational mechanisms that regulate gene expression [30], as a result of modifications in the chromatin conformation that allow arterial-venous-specific interactions among transcription factors at different promoter locations [7]. Seminal studies in systemic, umbilical, and placental EC focused on the endothelial nitric oxide synthase gene promoter suggest a comparable epigenetic profile among arterial EC [31,32,33,34]. Recently, we reported that transcriptional differences between HUAEC and HUVEC are also related to specific non-coding RNA/messenger RNA interactions [8]. In this study, we found that expression of arterial-venous markers is partially correlated with gene- and site-specific DNA methylation levels, supporting the role of epigenetic mechanisms in defining placental endothelium phenotypes. Furthermore, selected epigenetic markers showed a consistent capacity to recognize arterial (i.e., PLAEC) and venous (i.e., PLVEC, HUVEC) EC, suggesting their utility as biomarkers considering the relative stability and quantitative tools for assessing DNA methylation [35]. Conversely, we showed extensive transcriptional and DNA methylation differences among fetal, umbilical, and placental vessels, adding further evidence to the role of epigenetic mechanisms in endothelial heterogeneity. Endothelial cells from conduction vessels (e.g., HUVEC and HUAEC) are commonly used experimentally for in vitro cultures to model transport processes [36,37,38]; however, our results call for caution in extrapolating observations for these functions in these models. The massive differences in transcriptomic and epigenomic profiles point to developmental and functional differences, consistent with expected functions of blood vessels at different vascular levels of the umbilical-placental bed [1,2]. Notably, our findings concerning promoter- and DNA methylation-specific enrichment analysis, suggest the contribution of epigenetic mechanisms by affecting transcription factors related with key endothelial functions [39,40,41]. We propose that preparations from placental EC may be more appropriate for functional studies dealing with transport functions but keeping in mind that the preserved arterial-venous nature among microvascular cells may influence their behavior and equal representation in vitro [14].

As a limitation, this study compared datasets coming from different experimental conditions and platforms, which have a huge impact on variability. For instance, sex was not consistently reported in several datasets, not being possible to address sexual dimorphisms among endothelial cells. In contrast, all the comparisons were performed considering the experimental source (i.e., GSE) as a confounding factor, and including in the comparison cell type as a contrast factor, in addition to vascular level (i.e., large vs. small). Conversely, PLAEC and PLVEC samples come from the same dataset, sharing experimental, platform, and batch conditions, and also showed considerable variability when they were compared according to vascular level, in both, PCA and PHATE analysis. This was also evident in HUVEC analysis, which included data from different experimental datasets but clustered and grouped in finite distributions. This suggests that batch differences contribute to transcriptional variability among the samples to a lower degree compared to cell type and vascular level. Additionally, this study did not provide data on the DNA methylation profile in HUAEC using the Illumina platform available for PLAEC, PLVEC, and HUVEC. To partially solve this issue, epigenomic data from pulmonary artery EC (HPAEC) was used, as a proxy of large arterial fetal vessels. Nonetheless, recent evidence supports a tight correlation between transcriptional and epigenomic profiles among large artery endothelial cells, which also reflects the phenotypic differences with HUVEC [7,30]. Conversely, PLAEC and PLVEC data are derived from cultured EC isolated by chorionic perfusion, which may display a different profile from in vivo phenotypes, and be influenced by the mixed representation of resistance and microvascular cells. Further studies applying single-cell methodologies in freshly isolated tissue are required to confirm the occurrence of arterial-venous differences in small placental vessels, traits that are also observed in capillaries in systemic circulation since fetal development [4,5].

Altogether, this meta-analysis provides cogent evidence regarding the transcriptional and epigenomic profiles that differentiate among EC, proposing novel markers to define phenotypes based on micro- and macro-vascular levels. Moreover, these observations contribute new insights concerning endothelial heterogeneity within umbilical-placental beds with potential applications to correctly address the characterization of EC in health and diseased pregnancies.

## 4. Methods

### 4.1. Transcriptional Profiling

Transcriptomic datasets were selected, as previously described [8], applying guidelines from Prisma Equator for meta-analysis studies (www.equator-network.org/reporting-guidelines/prisma/, accessed on 2 August 2021). We browsed the Gene Expression Omnibus (www.ncbi.nlm.nih.gov/geo/, accessed on 25 August 2021) and OmicsDI (www.omicsdi.org, accessed on 25 August 2021) using the keywords “placental endothelial”, “PLAEC”, or “PLVEC” in combination with “Affymetrix”. Metadata, including cell source and culture conditions, were obtained for further analysis. The search considered all data available up to August 2021, retrieving 8 PLAEC and 8 PLVEC independent samples from a single dataset (GSE103552) and platform (ID: GPL6244). This dataset contained 37 samples, but only controls (non-diabetic) were considered for analysis, an exclusion criterion that was also applied for all the analyses in which only non-treated and/or control cells were considered. For HUAEC and HUVEC data, 68 previously validated reports were used [8] (HUAEC = 17; HUVEC = 51, platform ID GPL570, GSE codes detailed in Vega-Tapia et al., 2021). Discovery and cross-validation of microvascular gene markers were assayed by analyzing datasets of endothelial samples from diverse vascular beds (i.e., aorta, saphenous and umbilical vein (large vessels, GSE137041), and pulmonary and brain (microvascular, GSE73341) EC) (platform ID GPL6244). Raw data contained in CEL files were analyzed using the software Transcriptome Analysis Console 4.0.2.15 (Applied Biosystems) [8], considering cell type and vascular level as comparison parameters, HUVEC as the reference group, and GSE accession number as a variable factor to correct for batch and sample type effects. Analysis parameters considered the following cut-off values: −1.5 < fold change > 1.5; a *p*-value < 0.05 and FDR < 0.05 for differentially expressed genes (DEG) using the Limma package with empirical Bayes method for multiple comparison. Cross-platform analysis of DEG was performed using XLSTAT 2021.4 Life Sciences (Addinsoft), applying the following cut-off values: −1.5 < fold change > 1.5; a *p*-value < 0.01 and FDR < 0.05.

### 4.2. DNA Methylation Profiling

Genome-wide DNA methylation datasets were selected following guidelines from Prisma Equator for meta-analysis studies (www.equator-network.org/reporting-guidelines/prisma/, accessed on 2 August 2021). The browsing process was performed in Gene Expression Omnibus (www.ncbi.nlm.nih.gov/geo/, accessed on 25 August 2021) and OmicsDI (www.omicsdi.org, accessed on 25 August 2021) using the keywords “HUVEC”, “HUAEC”, “fetal endothelial”, “placental endothelial”, “PLAEC” or “PLVEC” in combination with “DNA methylation” to exclude data generated with other assays, resulting in 26 independent samples which included HUVEC (*n* = 5), PLAEC (*n* = 9), PLVEC (*n* = 9), and fetal pulmonary artery (HPAEC, *n* = 3) EC (accession codes: GSE82234, GSE106099, GSE140079). Dataset GSE106099 contained 30 samples, including gestational diabetes mellitus which were excluded, resulting in 18 control samples. Datasets with raw data (i.e., idat files) available were used to apply quality controls, normalization, and batch effect algorithms during analysis. Quality control and analysis of EPIC arrays were performed using ShinyÉPICo [42], an R-based graphical pipeline that uses minfi for normalization, and limma for differentially methylated positions analysis. Differentially methylated probes (DMP) and regions (DMR) were determined by contrasting EC levels (micro- vs. macrovascular), considering arterio-venous nature as a covariate, excluding X and Y chromosome probes, and array slide as “donor variable”. Differences were calculated using eBayes *t*-test analysis, and DMP and DMR were considered when a *p*-value < 0.05 and FDR < 0.05 were found.

### 4.3. Functional Enrichment

Functional enrichment for transcriptional data was carried out based on DEG according to macro- and micro-vascular, and arterial and venous placental EC comparisons. Gene ontology enrichment analysis was performed in GeneOntology (GO) (http://geneontology.org/, accessed on 28 February 2022) searching in Biological Processes (BP) database. Pathway enrichment was performed with Enrichr using the Kyoto Encyclopedia of Genes and Genomes (KEGG) and Reactome databases. Terms were selected filtering by False Discovery Rate (FDR) adjusted *p*-value (*p* < 0.05) and their relationship with general vascular function and cell differentiation.

Functional enrichment for DNA methylation data was analyzed using methylGSA [43], an R-based graphical pipeline that corrects the bias of differences in DNA methylation by gene length and the relative position of probes. DMP derived from DNA methylation profiling analysis was tested using the Gene Ontology, KEGG, and Reactome databases, applying a cut-off minimal gene set of 50 and methylglm as algorithm [44]. Additionally, transcription factors enrichment was predicted by selecting DMP laying on promoter regions (i.e., excluding intergenic, coding, and 3′UTR).

### 4.4. Clustering and Dimensionality Reduction

The clustering of samples based on transcriptional data and tSNE dimensionality reduction was performed using the R-based web tool Morpheus (https://software.broadinstitute.org/morpheus, accessed on 28 February 2022). Briefly, samples were clustered by rows (sample) and columns (transcripts) using Euclidean distance. T-distributed Stochastic Neighbor Embedding (tSNE) was used to determine the similarity among samples in transcriptional terms [45], and samples were embedded in the first two axes of the tSNE space. Clustering of methylation data was performed with ShinyÉPICo using Euclidean distances and considering rows (sample) and columns (probes or regions).

Multidimensional scaling (MDS) was performed using XLSTAT 2021.4 Life Sciences (Addinsoft) by a dissimilarity analysis with Euclidean distances from which the first two dimensions were derived to analyze the differences [45] among samples in transcriptional terms.

Finally, the Potential of Heat-diffusion for Affinity-based Trajectory Embedding (PHATE) was performed as an additional dimensionality reduction approach using the R implementation. PHATE squeezes all the variance of the high-dimensional manifold into a two-dimensional space, in which the samples are embedded [46].

### 4.5. Discovery and Validation of Molecular Markers

The correlation between gene expression and CpG methylation (probes) was evaluated with a correlation matrix of micro-macrovascular and arterial-venous DEG. Transcriptional and methylation data were paired by using samples coming from two related GSE datasets (GSE103552 and GSE106099). Highly correlated gene/probes, defined by *p* < 0.01 and an absolute r > 0.4 (Pearson correlation), were further filtered as predictors of microvascular or arterial EC by least absolute shrinkage and selection operator (Lasso) regression (discovery). Selected variables (i.e., probes) from Lasso analysis were then validated by multivariable logistic regression.

### 4.6. Statistical Analysis

All statistical analyses, tools, and software used in this study are described, when appropriate, along with each specific method detailed above.

### 4.7. Data Availability

The datasets analyzed in this study are mentioned in Section 4. All accession details and download links are described in Appendix A.

### 4.8. Code Availability

No custom codes were generated to be used in this data analysis. Software used and accession is detailed in Section 4.

## Figures and Tables

**Figure 1 ijms-23-11873-f001:**
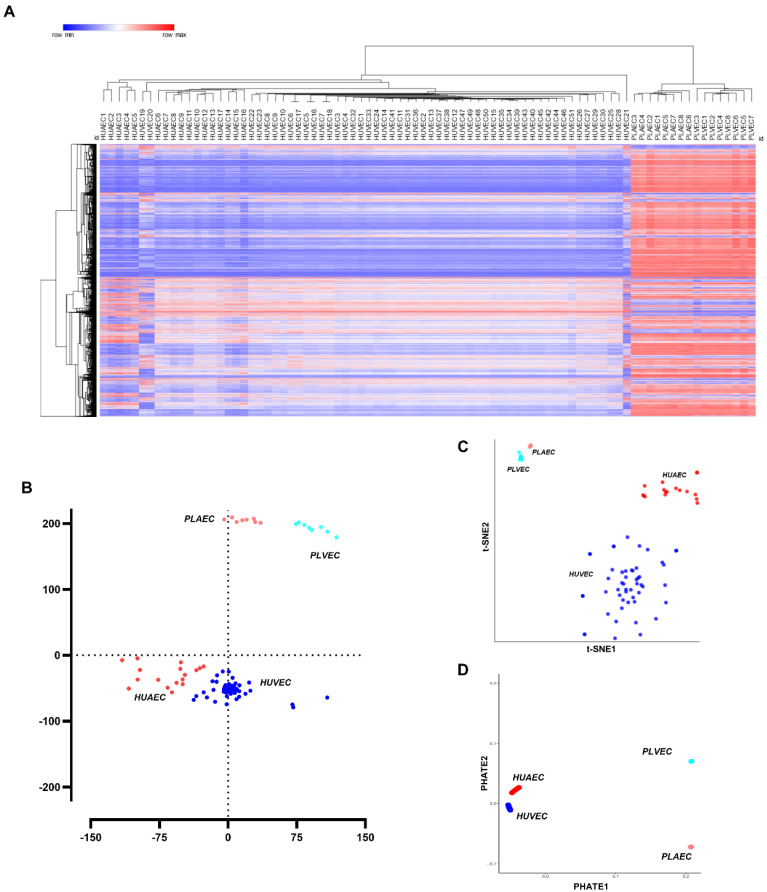
Transcriptional profiling, grouping, and proximity of umbilical and placental endothelial cells. (**A**) Hierarchical analysis of 17,280 transcripts commonly expressed in endothelial cells of the umbilical-placental bed. MDS (**B**), t-SNE (**C**), and PHATE (**D**) plots showing dissimilarities, similarities, and relative proximity among HUAEC (red dots), HUVEC (blue dots), PLAEC (light red dots), and PLVEC (light blue dots).

**Figure 2 ijms-23-11873-f002:**
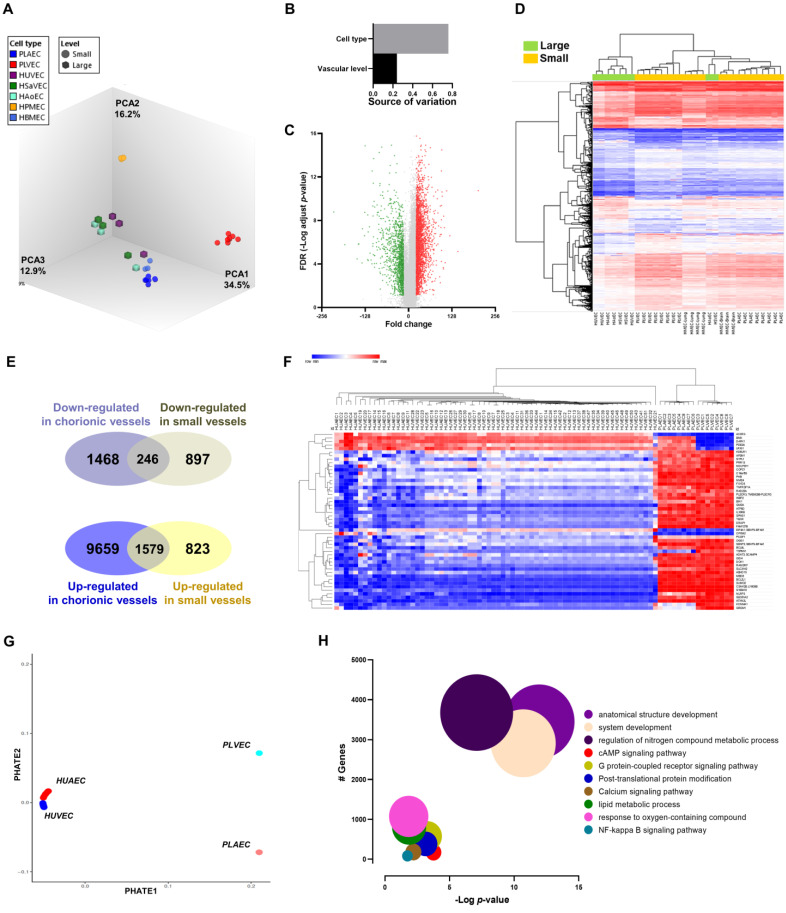
Transcriptional profile of macro and microvascular endothelial cells. (**A**) PCA analysis of samples distinguishing the groups by the organ of origin (color) and level (shape, circle, or square) and specifying the number of macro- (*n* = 9) and microvascular samples (*n* = 23) with the respective filter criteria and pie chart with upregulated (4.45%) and downregulated (9.78%) genes between samples. (**B**) Graph bar showing the contribution of cell type and vascular level origin (i.e. large or small vessels) to transcriptional variation (**C**) Volcano plot showing 1295 downregulated genes (green points) and 2847 upregulated genes (red points) comparing small vs large vessels with respective fold changes (**D**) Level hierarchical clustering between macrovascular (green) and microvascular (yellow) samples. (**E**) Venn diagram showing the overlap between downregulated (left) and upregulated (right) genes between chorionic and microvascular samples. (**F**) Hierarchical analysis of top 50 DEGs between macro-and micro-vascular endothelial cells. (**G**) PHATE plot depicts proximity between macro- and micro-vascular samples considering top 50 DEGs. (**H**) Bubble plot shows the enriched BP and pathways for differentially expressed genes between micro and macrovascular placental endothelial cells. Bubble size is representative of the number of genes regulated for the given term.

**Figure 3 ijms-23-11873-f003:**
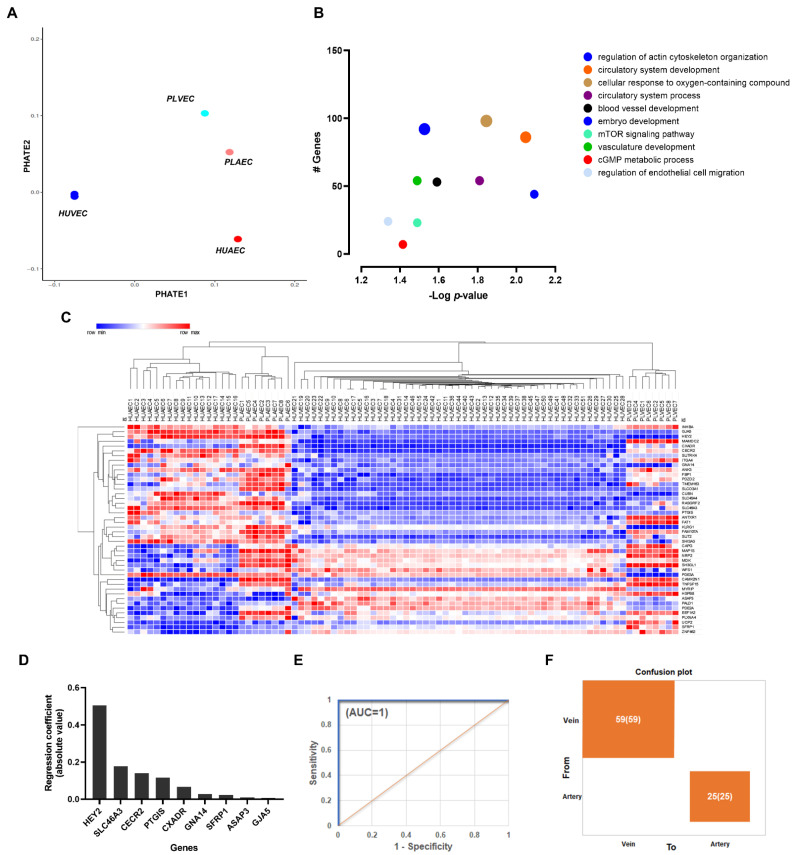
Transcriptional markers of arterial-venous identity for umbilical and placental vessels. (**A**) The PHATE plot shows the proximity between endothelial cells samples considering arterial-venous markers. (**B**) Bubble plots show the enriched BP and pathways for differentially expressed genes between arterial and venous endothelial cells. Bubble size represents the number of genes regulated for each term. (**C**) Hierarchical clustering of arterial-venous DEGs in umbilical and placental endothelial cells. (**D**) Lasso regression analysis addressed 9 genes highly correlated with arterio-venous nature dichotomy. (**E**) Logistic regression and (**F**) confusion matrix plot validates the high discriminating ability for 9 DEGs.

**Figure 4 ijms-23-11873-f004:**
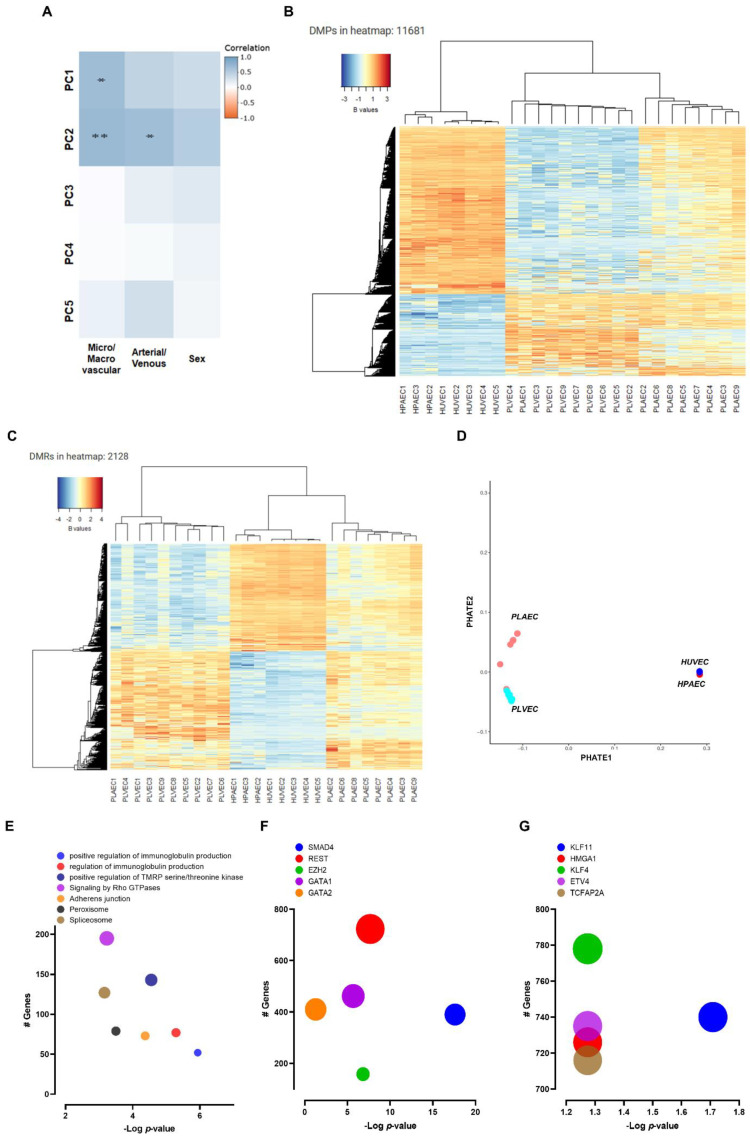
DNA methylation profile of micro- and macro-vascular endothelial cells. (**A**) PCA analysis for the first five principal components for micro- and macro-vascular, arterio-venous, and sex-differentiated samples. (**B**) Heatmap for DMPs and DMRs (**C**) in each sample. (**D**) PHATE proximity plot to each sample: HPAEC (red dots), HUVEC (blue dots), PLAEC (light red dots), and PLVEC (light blue dots). (**E**) Epigenomic-specific functional enrichment analysis bubble plots depict the enriched pathways and transcriptions factor generated by ENCODE (**F**) and TRANSFAC and JASPAR databases (**G**). Bubble size represents the number of genes regulated for each term. In A, * *p* < 0.05, ** *p* < 0.01 *p*-values for correlation between principal components and sample traits.

**Figure 5 ijms-23-11873-f005:**
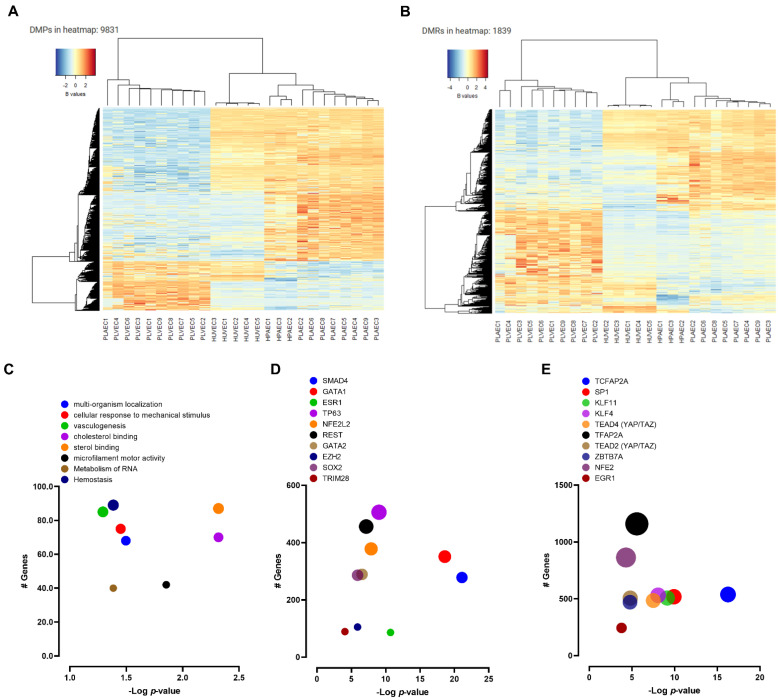
DNA methylation profile of arterial and venous endothelial cells. (**A**) Heatmap for DMPs and DMRs (**B**) in each sample. (**C**) Epigenomic-specific functional enrichment analysis bubble plots depict the enriched pathways and transcriptions factor generated by ENCODE (**D**) and TRANSFAC and JASPAR databases (**E**).

**Figure 6 ijms-23-11873-f006:**
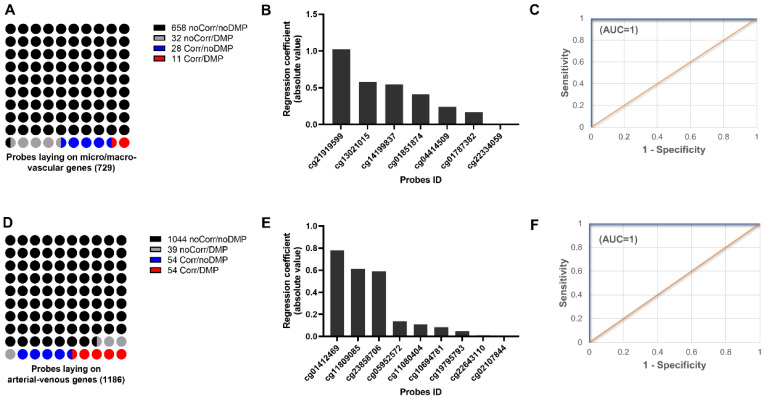
Validation of DEG/DMC for arterial-venous and umbilical/placental identity. (**A**) Schematic proportional representation of probes laying micro- and macro-vascular genes. Proportions are represented by no correlated and no DMP genes (black dots), no correlated but DMP genes (gray dots), correlated but no DMP genes (red dots), and correlated and DMP genes (red dots). (**B**) Graph bars show six probes ID with high predictive power differentiating microvascular endothelial cells tested by Lasso regression analysis and validated by logistic regression (**C**). (**D**) Schematic proportional representation of probes laying arterio-venous genes. Proportions are represented by no correlated and no DMP genes (black dots), no correlated but DMP genes (gray dots), correlated but no DMP genes (red dots), and correlated and DMP genes (red dots). (**E**) Graph bars show nine probes ID with high predictive power differentiating arterio-venous nature of endothelial cells tested by Lasso regression analysis and validated by logistic regression (**F**).

**Table 1 ijms-23-11873-t001:** Infinium probes with high DNA methylation/transcript level correlation among microvascular markers.

Gene	Probe ID	r2	*p*-Value
ACKR3	cg27529004	0.5049	0.002
	cg26960322	0.4845	0.0027
	cg27529004	0.4179	0.0068
ATP5D	cg25732045	0.5784	0.0006
C14orf2	cg24129873	0.4053	0.008
CFH	cg23557926	0.4591	0.0039
DCXR	cg07847925	0.4022	0.0083
MBD6	cg09687907	0.402	0.0084
HAS3	cg00451194	0.6541	0.0001
	cg02562299	0.6135	0.0003
	cg02730714	0.5877	0.0005
	cg12108730	0.5639	0.0008
	cg26818159	0.5668	0.0008
MBD6	cg09687907	0.402	0.0084
MYH10	cg08493106	0.501	0.0022
	cg23163754	0.5323	0.0013
MECOM	cg01787382	0.6061	0.0004
	cg04414509	0.597	0.0005
	cg12847986	0.4631	0.0037
	cg13021015	0.4572	0.004
	cg20201475	0.6523	0.0002
	cg20203114	0.4009	0.0085
NLRP3	cg11422335	0.5182	0.0017
	cg18183941	0.4599	0.0039
	cg18793688	0.488	0.0026
	cg21919599	0.4156	0.007
PCGF4	cg18826743	0.3991	0.0087
PLSCR3	cg13775913	0.4792	0.003
RUNX1T1	cg00045118	0.4515	0.0044
	cg03760919	0.556	0.0009
	cg07120544	0.5728	0.0007
	cg18457433	0.5135	0.0018
	cg22334059	0.6667	0.0001
SENP3	cg01733795	0.5143	0.0018
VPS72	cg14199837	0.5746	0.0007
	cg22513511	0.5242	0.0015
	cg24694326	0.6687	0.0001
WBP2	cg19824334	0.5601	0.0009
ZNF561	cg23227837	0.5248	0.0015

**Table 2 ijms-23-11873-t002:** Infinium probes with high DNA methylation/transcript level correlation among arterial-venous markers.

Gene	Probe ID	r2	*p*-Value
ANTXR1	cg00240205	0.6279	0.0003
	cg10803714	0.7891	0.0001
ASAP3	cg20847090	0.5708	0.0007
CUBN	cg11298524	0.4825	0.0028
	cg17436460	0.4577	0.004
CXADR	cg20541233	0.5772	0.0006
EEF1A2	cg10786876	0.4521	0.0043
	cg11080404	0.4124	0.0073
	cg23858706	0.5094	0.0019
FAT1	cg01337940	0.4239	0.0063
	cg01454936	0.571	0.0007
	cg01585180	0.5632	0.0008
	cg02325160	0.487	0.0026
	cg02596645	0.4254	0.0062
	cg02998591	0.5382	0.0012
	cg03844826	0.5928	0.0005
	cg08100122	0.4206	0.0066
	cg08288016	0.4092	0.0076
	cg11970085	0.3911	0.0096
	cg12876620	0.476	0.0031
	cg15374133	0.6197	0.0003
	cg15815948	0.4592	0.0039
	cg17755082	0.6884	0.0001
	cg24792658	0.3975	0.0088
	cg24820270	0.4635	0.0037
	cg24890045	0.6032	0.0004
FBP1	cg01210663	0.5108	0.0019
	cg13982688	0.4929	0.0024
GJA5	cg02103822	0.4502	0.0044
	cg27546670	0.404	0.0081
	cg00421368	0.4832	0.0028
	cg06617692	0.4726	0.0033
	cg14146751	0.4954	0.0023
	cg14232851	0.4072	0.0078
GNA14	cg14232851	0.4519	0.0043
HEY2	cg00066750	0.5169	0.0017
	cg22060817	0.6908	0.0001
HSPB8	cg11187110	0.5173	0.0017
	cg16425829	0.6418	0.0002
	cg24694702	0.5929	0.0005
	cg05361406	0.5221	0.0016
INHBA	cg01412469	0.6064	0.0004
ITGA4	cg16057262	0.476	0.0031
	cg20415809	0.407	0.0078
KLRG1	cg00443307	0.7357	0.0001
	cg14913610	0.7502	0.0001
	cg26806779	0.8603	0.0001
MAMDC2	cg13870494	0.583	0.0006
	cg14649449	0.5675	0.0008
MYRIP	cg23024967	0.3905	0.0096
NRP2	cg05348875	0.4383	0.0052
	cg14157435	0.4339	0.0055
	cg01154445	0.4072	0.0078
	cg19795793	0.3987	0.0087
	cg26422981	0.3942	0.0092
PALD1	cg01464515	0.5916	0.0005
	cg14683490	0.4362	0.0054
	cg20418394	0.391	0.0096
	cg22643110	0.6028	0.0004
PDE2A	cg05952572	0.4508	0.0044
PDE3A	cg04724646	0.4541	0.0042
	cg12136731	0.816	0.0001
	cg13063900	0.6577	0.0001
	cg13446110	0.6396	0.0002
	cg18639524	0.3884	0.0099
	cg21913301	0.7737	0.0001
	cg23015991	0.6685	0.0001
	cg24336686	0.5668	0.0008
PLXNA4	cg04126760	0.4102	0.0075
	cg10694781	0.5089	0.0019
RASGRF2	cg09571630	0.394	0.0092
	cg15341601	0.4573	0.004
SFRP1	cg00000321	0.4361	0.0054
	cg06166767	0.4027	0.0083
SH3GL1	cg11592634	0.4155	0.007
	cg22305516	0.5871	0.0005
SHISA3	cg11065575	0.4044	0.0081
SLC45A4	cg01081737	0.6167	0.0003
	cg07769015	0.4194	0.0067
	cg15586392	0.5482	0.001
SLCO3A1	cg00314343	0.6501	0.0002
	cg01240823	0.5338	0.0013
	cg02107844	0.5093	0.0019
	cg02633036	0.4318	0.0057
	cg03756778	0.5045	0.002
	cg04200243	0.5267	0.0015
	cg07725123	0.5644	0.0008
	cg11330307	0.6864	0.0001
	cg11680590	0.5256	0.0015
	cg11859489	0.4483	0.0046
	cg12034869	0.5715	0.0007
	cg12667732	0.7015	0.0001
	cg13039199	0.5038	0.0021
	cg17266475	0.3995	0.0086
	cg19621065	0.545	0.0011
	cg19687529	0.4041	0.0081
	cg20464948	0.3904	0.0097
	cg21656246	0.4978	0.0023
	cg21715599	0.4385	0.0052
	cg26467725	0.3902	0.0097
	cg26677406	0.5366	0.0013
	cg27070951	0.6864	0.0001
SLIT2	cg07290920	0.4456	0.0047
TMEM163	cg17319374	0.4106	0.0075
TNFSF15	cg10791260	0.4555	0.0041
	cg11809085	0.7195	0.0001
ZNF462	cg13635022	0.3988	0.0087
	cg13576904	0.4373	0.0053

## Data Availability

The datasets analyzed in this study are mentioned in Section 4. All accession details and download links are described in Appendix A.

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
