# Peer review of "Transcriptional and Epigenomic Markers of the Arterial-Venous and Micro/Macro-Vascular Endothelial Heterogeneity within the Umbilical-Placental Bed"

_ijms, 2022, doi:10.3390/ijms231911873_

Round 1

Reviewer 1 Report

In this study, the authors performed a variety of in silico analyses using transcriptomic and epigenetic data obtained from publicly available databases and from a wide variety of studies. Samples analyzed ranged from placental and umbilical endothelial cells to adult arterial and venous endothelial cells from various blood vessel types. In addition, some samples are from cultured cells, others from fresh tissue.

It is very difficult/impossible to find out the characteristics of the samples which data were used in the study. Supplementary Table 1 lists all the accession number of the data, but, a more detailed description would be welcome (how many samples, tissue or cell culture, sex, platform/technology used, etc).  

I am mostly concerned by the fact that the authors compare transcriptomic data coming from various sources (placenta/umbilical samples, cultured cells from various tissues) obtained from various platform types (different sets of microarrays). The clustering of the samples mostly reflects sample type, which was expected, but could also reflect differences in the way the data were obtained or sample characteristics (sampling mode, individual age/sex, etc). The authors should explain how they cope with batch/platform/sample type effect.

The differences in the level of gene expression between different cell origin and type (placenta vs umbilical, artery vs vein, new born vs adult) are obviously large enough to overcome the variability due to the origin and variety of the data/sample/platform. However, this should certainly be addressed in the analysis and the interpretation.

Table 1: the authors performed correlation analysis between DNA methylation and transcript expression levels of several genes that were shown to be microvascular specific markers. However, because the gene expression and methylation data were obtained from separate samples and datasets, they must explain how they performed the correlation and justify the relevance and adequacy of their analysis.

Author Response

Comments and Suggestions for Authors – Reviewer 1

In this study, the authors performed a variety of in silico analyses using transcriptomic and epigenetic data obtained from publicly available databases and from a wide variety of studies. Samples analyzed ranged from placental and umbilical endothelial cells to adult arterial and venous endothelial cells from various blood vessel types. In addition, some samples are from cultured cells, others from fresh tissue.

  1. It is very difficult/impossible to find out the characteristics of the samples which data were used in the study. Supplementary Table 1 lists all the accession number of the data, but, a more detailed description would be welcome (how many samples, tissue or cell culture, sex, platform/technology used, etc).

Reply 1. To improve the details about each dataset used, an updated version of Supplementary table 5, including a description of the datasets is now included. Also, some commentaries across the manuscript addressing the points raised were included.

  1. I am mostly concerned by the fact that the authors compare transcriptomic data coming from various sources (placenta/umbilical samples, cultured cells from various tissues) obtained from various platform types (different sets of microarrays). The clustering of the samples mostly reflects sample type, which was expected, but could also reflect differences in the way the data were obtained or sample characteristics (sampling mode, individual age/sex, etc). The authors should explain how they cope with batch/platform/sample type effect.
  2. The differences in the level of gene expression between different cell origin and type (placenta vs umbilical, artery vs vein, new born vs adult) are obviously large enough to overcome the variability due to the origin and variety of the data/sample/platform. However, this should certainly be addressed in the analysis and the interpretation.

Reply 2 & 3. We agree with the reviewer concerning the effect that different experimental conditions and platforms have a huge impact on variability. To address this issue, all the comparisons were performed considering the experimental source (i.e. GSE) as a confounding factor, and including in the comparison cell type as a contrast factor, in addition to vascular level (i.e. large vs small). Further details on this have been included in “Methods - Transcriptional profiling”.

Concerning sexual dimorphism, unfortunately, due to the lack of consistency in the sex data among the different datasets analyzed, that factor cannot be addressed in this work. A brief comment was added in “limitations”.

Considering this background, it is worth noting that PLAEC and PLVEC samples come from the same dataset, sharing experimental, platform and batch conditions, but they showed considerable variability when they were compared according to vascular level, in both, PCA and PHATE analysis. This was also evident in HUVEC analysis, which come from different experimental datasets, but clustered and grouped in finite distributions. This suggests that batch differences contribute to a lesser degree to transcriptional variability among the samples, compared to cell type and vascular level. A brief comment on this was also included in the limitation paragraph (discussion).

  1. Table 1: the authors performed correlation analysis between DNA methylation and transcript expression levels of several genes that were shown to be microvascular specific markers. However, because the gene expression and methylation data were obtained from separate samples and datasets, they must explain how they performed the correlation and justify the relevance and adequacy of their analysis.

Reply 4. We apologize for the lack of clarity concerning this part of the analysis. The dataset used for transcriptional (GSE103552) and DNA methylation (GSE106099) are paired: they include the same samples analyzed for transcript levels and gene methylation, which allows to derived precise expression-methylation correlations when samples are paired. To address this issue, details were briefly included in Results (Validation of epigenetic markers for fetal, umbilical, and placental EC heterogeneity) and Methods (Discovery and validation of molecular markers). 

Reviewer 2 Report

In this manuscript entitled “Transcriptional and epigenomic markers of the arterial-venous and micro/macro-vascular endothelial heterogeneity within the umbilical-placental bed”, the authors characterize the transcriptional profile of HUAEC, PLAEC, PLVEC, and HUVEC. In particular, the authors hypothesized the contribution of epigenetic mechanisms and analyzed whether they affect transcription factors associated with key endothelial functions.

Overall, the observations of this study are interesting. Most of the data in this manuscript are compelling and are presented through well-designed experiments. 

Comments:

1. In Fig. 6 E, the authors find probes that correlate both methylation and gene expression. For example, have INHBA (cg01412469) and TNFSF15 (cg11809085) actually been reported to methylate? And if these proteins may be involved in epigenetic regulation, what are the possible mechanisms?

Author Response

Comments and Suggestions for Authors – Reviewer 2

 In this manuscript entitled “Transcriptional and epigenomic markers of the arterial-venous and micro/macro-vascular endothelial heterogeneity within the umbilical-placental bed”, the authors characterize the transcriptional profile of HUAEC, PLAEC, PLVEC, and HUVEC. In particular, the authors hypothesized the contribution of epigenetic mechanisms and analyzed whether they affect transcription factors associated with key endothelial functions.

Overall, the observations of this study are interesting. Most of the data in this manuscript are compelling and are presented through well-designed experiments. 

Comments:

  1. In Fig. 6 E, the authors find probes that correlate both methylation and gene expression. For example, have INHBA (cg01412469) and TNFSF15 (cg11809085) been reported to methylate? And if these proteins may be involved in epigenetic regulation, what are the possible mechanisms?

Reply. We thank the reviewer for raising this interesting possibility, which we didn´t address in the original version of the manuscript. In our analysis, we only tested for a correlation between methylation status in the gene locus (as indicated by differentially methylated probes) and gene expression (by transcript levels). The genes selected by LASSO are indeed methylated in the datasets analyzed, but they don´t are involved themselves in epigenetic regulation. These genes participate in a range of processes, including cell migration and solute transport; these processes are important for endothelial cell function, consistent with the endothelial nature of the samples. We now include this information in the new version of the manuscript. In future studies, it will be highly interesting to explore directly whether bona fide epigenetic regulators are differentially regulated in endothelial cells of distinct origins in the umbilical-placental beds; and to analyze their potential roles in transcriptional differences between them.